# Towards single-stage prediction of vertebrae segmentation for opportunistic screening of 3D CT scans

**Hellena Hempe**                                          HELLENA.HEMPE@UNI-LUEBECK.DE
**Mattias P. Heinrich**                                        HEINRICH@IMI.UNI-LUEBECK.DE
*Institute of Medical Informatics, Universität zu Lübeck, Lübeck, Germany*

## Abstract

Opportunistic screening of the spine using routine CT calls for light-weight localisation and segmentation networks. We explore the performance of a single-stage network based on the MobileNetV3 and the DeepLab/ASPP on the VerSe20 Challenge dataset. Our proposed method improves the robustness of localisation and segmentation in comparison to other design choices of network architectures.

**Keywords:** Spine CT, Semantic segmentation, Localisation, Light-weight, Efficiency

## 1. Introduction

Opportunistic screening of CT acquired in clinical practice can be used for early diagnosis of osteoporosis and other degenerative diseases of the spine. In a resource limited clinical setting, light-weight networks that perform localisation and segmentation of each vertebrae are needed. The VerSe19 and '20 challenges aim at addressing the task of spine CT analysis but most proposed methods (Sekuboyina et al., 2020) require multiple-stages and are computationally complex. Our approach aims to reduce the computational complexity of the labelling task while improving the robustness in order to facilitate the application for opportunistic screenings. Therefore, we propose a light-weight model that combines the efficiency of the MobileNetV3 (Howard et al., 2019) with Atrous Spatial Pyramid Pooling (ASPP) (Chen et al., 2018).

## 2. Methods

To improve our networks efficiency, we applied the idea of compound scaling (Tan and Le, 2019). We experimented the networks depth, width and resolution searching for the best outcome within 2hours training time. Our model consists of three components: a backbone model, the ASPP and a segmentation head with a single skip connection to the backbone. The backbone comprises 8 efficient blocks similar to the MobileNetV3 (Howard et al., 2019) with 24 layers in total. We used strides in the first layer and in the middle layer of the 6th block to lower the resolution and increase network width for deeper layers. The segmentation head predicts 29 classes, for 28 vertebrae labels of the dataset.

---

\* The authors sincerely acknowledge the ARTEMIS consortium

## 3. Experiments and Results

Our method was trained on the VerSe20 training dataset (Liebl et al., 2021) [123] with a 3-fold split for validation. The images were preprocessed by normalising the orientations and gray values, and donwsampling to 128x128x128. Training was performed with random size 4 mini-batches, using the ADAM optimizer with learning rate 0.001. We employed data augmentation by performing affine grid transformations. As loss function, we used the NLLLoss with class weights in combination with an Edge-Loss (Ronneberger et al., 2015). With our learning configuration we could stop after 1500 iterations, which took less than 1h30min on a NVIDIA RTX 2080 Ti GPU. We compare our method to other segmentation networks such as the U-Net (Ronneberger et al., 2015) and the ResNet18 (He et al., 2016) in combination with ASPP, using the same learning strategy. The nnUNet framework (Isensee et al., 2021) in 3D full resolution mode was trained as a baseline. The results show that our proposed method outperforms the other networks when comparing Dice scores while reducing the number parameters see Table 1. Figure 1 shows our plots for evaluating the localisation accuracy and an example of a predicted segmentation. The plots show the labels of each vertebrae at the centre-mass of the height. Each color represents one CT scan of the 21 validation cases.

Table 1: Mean dice scores and trainable parameters after number of epochs for each method

| Method | Multi-label Dice | Binary Dice | Parameters | Epochs |
|---|---|---|---|---|
| ResNet18/ASPP | 23.2% | 67.1% | 3.2M | 1500 |
| U-Net | 36.9% | 73.8% | 5.2M | 1500 |
| MobileNet | 19.4% | 67.1% | 277k | 1500 |
| MobileNet/ASPP with flips | 45.0% | 71.0% | 1.2M | 1500 |
| MobileNet/ASPP | 53.8% | 77.9% | 1.2M | 1500 |
| nnUNet with flips | 42.8% | – | – | 200 |

## 4. Discussion

Our approach improves the robustness of vertebrae labelling and can be viewed as a lightweight alternative to existing approaches. The ASPP contributes to more robust labelling results, as well as emphasising edges using edge-loss. Our method still has potential to optimize by increasing the number of trained epochs. Our network labelled cropped vertebrae additionally, but was more robust against one-label-shifts and misclassification than other methods. Applying flips destabilized the training results which can be seen in the nnUNet results, therefore we show our results without flips for data augmentation. This could be due to the anatomical structure of the spine and needs further investigation. We believe that our method shows promising results as a first step to solve the vertebrae localisation and

1. (Sekuboyina et al., 2020)
2. M. T. Löffler, A. Sekuboyina, and A. Jacob et al. A vertebral segmentation dataset withfracture grading.Radiology: Artificial Intelligence, 2(4):e190138, 2020.
3. B. Glocker, J. Feulner, and A. Criminisi et al. Automatic localization and identification ofvertebrae in arbitrary field-of-view ct scans. MICCAI, pages 590–pp. Springer, 2012.

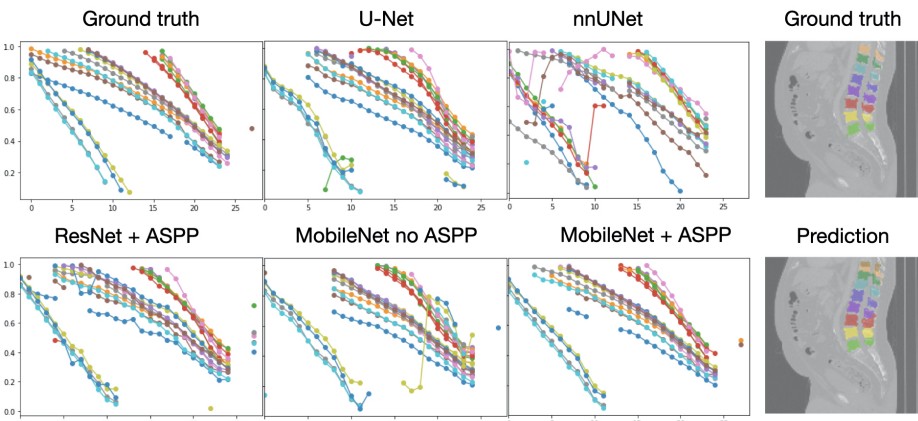

Figure 1: Localisation outcome of different network architectures and an example segmentation from our best model

segmentation task and we aim towards a light-weight single-stage network for this purpose in the future. As a next step, the model could be trained within the nnUNet framework.

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
