# OpenReview forum: "Towards single-stage prediction of vertebrae segmentation for opportunistic screening of 3D CT scans"
_MIDL.io/2021/Conference/Short — Submitted to MIDL 2021_

### Official Review · Reviewer_6mev · 2021-04-25

**Confidence:** 4
**Final Rating:** 2

**Summary:**

The authors target an important clinical problem (spine screening) using a light-weight localization and segmentation network based on recent developments in efficient architectures, such as MobileNetV3 and ASPP. The authors find that flipping as data augmentation step destabilizes network training and propose good suggestions for the next steps.

**Strengths:**

The motivation is clear and the results are promising for medical usage. The use of different networks and incorporation of latest neural networks (nnUnet) is promising. Code on Github is provided for reproducibility.

**Weaknesses:**

•	Network architectures. I am not sure what kind of layers of which MobileNet architecture were implemented. I thank the authors for providing code to answer this question. This provided code, however, shows classical MobileNetV2 layers as far as I can tell, hallmarks of MobileNetV3 are the Squeeze-and-Excite-layer in the residual layers and the use of the hard-swish activation function that I can’t observe in the provided code. Also the U-Net architecture that only contains 5.2M parameters is presumably adapted and not a re-implementation of the original network (around 22M parameters). Further, I am not sure how the MobileNet without ASPP is constructed to gain a segmentation mask.
•	Reasoning and comparison. The rational why ASPP was used and not any other decoder structure is missing, further a fair comparison would be to use different MobileNet backbones (V1,V2,V3) and different decoders structures would be reasonable. Knowing this is a short paper with preliminary data, I still believe that a more concise paper would strengthen the presented results. Discussing that longer training would be beneficial for the performance, but claiming that 1500 epochs (or iterations?) is sufficient is puzzling to this reviewer. I think the authors can improve their statements in a revised version.


**Deanonymize Review:**

no

**Detailed Comments:**

•	“Our method was trained on the VerSe20 training dataset (Liebl et al., 2021)123 with a 3-fold split for validation.” – citation is mixed with footnotes containing citation and references that should belong to the reference section
•	Figure 1 lacks unfortunately x- and y-axis labels and/or a detailed caption. The Figure content is explained in the text, though, but I think correct axes labels are important in a scientific paper.

**Justification Of The Rating:**

This reviewer is unsure about the correct implementations of the neural network architectures, and thinks besides the high potential of this work and general interest for MIDL, the paper as is has several weaknesses limiting its impact, but I would consider it as borderline.

**Paper Type:**

both

**Special Issue:**

no

---

### Official Review · Reviewer_ezuK · 2021-04-26

**Confidence:** 5
**Final Rating:** 2

**Summary:**

The paper describes a method for segmentation and labeling of the vertebrae in CT images. The method is based on MobileNetV3 in combination with Atrous Spatial Pyramid Pooling. The paper claims that this is a particularly light-weight deep learning model that might have benefits over more complex approaches when it comes to implementation in clinical practice. The paper uses the VerSe 2020 dataset for training and evaluation.

**Strengths:**

* A robust approach for developing efficient models that can perform tasks like vertebra segmentation and labeling in a short amount of time or on regular workstations would be of great use, the topic of this paper is relevant
* The proposed method is evaluated on a publicly available dataset, which allows for a direct comparison with other methods


**Weaknesses:**

While motivation and method are coherent, the experiments are quite disorganized. Since the introduction emphasizes the need for a light-weight and efficient solution, it is a bit surprising that the runtimes of the different approaches are not reported. Furthermore, the results of the nnUNet are included as a baseline but this model was only trained with 200 epochs without providing any explanation even though it normally trains for a fixed number of 1000 epochs. Also, the binary dice score is not reported for this model even though that should be trivial to compute, and the nnUNet model was trained with flips as a data augmentation strategy while the authors later state that this apparently destabilized the performance - the proposed model was therefore trained both with and without flips while the nnUNet was only trained with flips, which does not seem like a fair comparison. It is anyway a bit strange to compare with the nnUNet as the baseline rather than comparing with methods that participated in the VerSe 2020 challenge, for which the results are available online. Using the same metrics that were used in the challenge would have been beneficial for this.

Overall, the paper seems to make the point that the proposed method outperforms other network architectures while being more light-weight (lower number of parameters). However, the value of this finding is somewhat limited since none of the models in the comparison reached an acceptable performance. The majority of the method on the challenge leaderboard reached a higher dice score. How well the method works in terms of labeling the vertebrae is unfortunately hard to judge because the identification error is not reported. Inferring the identification error rate from the reported multi-class dice score is not really possible since this score is a mix of segmentation and labeling errors.

**Deanonymize Review:**

no

**Detailed Comments:**

The paper mentions that the networks were trained with 29 output classes since the Verse 2020 dataset contains 28 foreground labels - however, 2 of these labels are actually only reserved for future use, there are no examples of these classes in the dataset (sacrum and cocygis) so it does not really make sense to include them as output classes at this point.

It would be helpful if the description of Figure 1 would be moved from the main body to the caption of Figure 1. The plots are also missing axis labels.

**Justification Of The Rating:**

The paper proposes a novel combination of two well-known concepts, namely MobileNetV3 and ASPP. However, while this is an interesting approach with some benefits over more complex approaches, it does unfortunately not seem to work so well in terms of segmentation accuracy, and whether it holds any promise in terms of labeling accuracy is not clear from the presented results.

**Paper Type:**

methodological development

**Special Issue:**

no

---

### Meta-Review · Program_Chairs · 2021-05-11

**Recommendation:** Reject
**Confidence:** 5

**Metareview:**

Two reviewers share the same opinions. The authors are suggested to improve the work according to the raised comments.

---

### Decision · Program_Chairs · 2021-05-11

Reject